# PERFORMANCE HETEROGENEITY IN MESSAGE-PASSING AND TRANSFORMER-BASED GRAPH NEURAL NETWORKS

## ABSTRACT

Graph Neural Networks have emerged as the most popular architecture for graph-level learning, including graph classification and regression tasks, which frequently arise in areas such as biochemistry and drug discovery. Achieving good performance in practice requires careful model design. Due to gaps in our understanding of the relationship between model and data characteristics, this often requires manual architecture and hyperparameter tuning. This is particularly pronounced in graph-level tasks, due to much higher variation in the input data than in node-level tasks. To work towards closing these gaps, we begin with a systematic analysis of individual performance in graph-level tasks. Our results establish significant performance heterogeneity in both message-passing and transformer-based architectures. We then investigate the interplay of model and data characteristics as drivers of the observed heterogeneity. Our results suggest that graph topology alone cannot explain heterogeneity. Using the Tree Mover's Distance, which jointly evaluates topological and feature information, we establish a link between class-distance ratios and performance heterogeneity in graph classification. These insights motivate model and data preprocessing choices that account for heterogeneity between graphs. We propose a selective rewiring approach, which only targets graphs whose individual performance benefits from rewiring. We further show that the optimal network depth depends on the graph's spectrum, which motivates a heuristic for choosing the number of GNN layers. Our experiments demonstrate the utility of both design choices in practice.

## 1 INTRODUCTION

Graph Neural Networks (GNNs) have found widespread applications in the social, natural and engineering sciences (Zitnik et al., 2018; Wu et al., 2022; Shlomi et al., 2020). Notable examples include graph classification and regression tasks, which arise in drug discovery (Zitnik et al., 2018), protein function prediction (Gligorijević et al., 2021), and the study of chemical reactions (Jin et al., 2017; Coley et al., 2019), among others.

Most state of the art GNNs are based on message-passing or transformer-type architectures. In both cases, careful model design and parameter choices are crucial for competitive performance in downstream tasks. A growing body of literature studies the relationship of model and data characteristics in graph learning. This includes the study of challenges in encoding long-range dependencies, which arise in shallow architectures due to under-reaching (Barceló et al., 2020) and in deep architectures due to over-smoothing and over-squashing effects (Alon & Yahav, 2021; Li et al., 2018). A second perspective evaluates a model's (in)ability to encode certain structural functions due to limitations in representational power (Xu et al., 2018). Some recent works have studied model and data characteristics through classical complexity lenses, such as generalization (Garg et al., 2020; Le & Jegelka, 2024; Franks et al., 2024) and trainability (Kiani et al., 2024). While these results offer valuable theoretical insights, their ability to directly guide design choices for specific datasets and tasks is often limited. As a result, model design usually relies on manual hyperparameter tuning in practise.

In this work, we study the interplay of model and data characteristics from a different perspective. We analyze the performance of a model on individual graphs with the goal of understanding varia-

tions in optimal model design within data sets. We introduce *heterogeneity profiles* as a tool for systematically evaluating individual performance across graphs in a dataset. The analysis of the profiles of several classification and regression benchmarks for both message-passing and transformer-based architectures reveals significant performance heterogeneity in graph-level learning. We then investigate how data and model characteristics drive this heterogeneity. A natural idea in this context is that the topological properties of the graphs, along with their variation across the dataset, might explain the observed heterogeneity. However, our results indicate that graph topology alone cannot explain heterogeneity. We then analyze the datasets with the Tree Mover's Distance (Chuang & Jegelka, 2022), a similarity measure that compares graphs using both topological and feature information. Using this lens, we show that common graph-level classification benchmarks contain examples that are more "similar" to graphs of a different label than to graphs with the same label. The prediction stability of typical GNN architectures therefore makes it hard to classify these examples correctly.

Using these insights, we study data pre-processing and model design choices with an eye towards heterogeneous graph-level effects. We first revisit graph rewiring, a pre-processing technique that perturbs the edges of input graphs with the goal of mitigating over-smoothing and over-squashing. We find that while some graphs benefit, the individual performance of others drops significantly as a result of rewiring. Although we observe notable differences between rewiring techniques, the general observation is consistent across approaches and data sets. Motivated by this observation, we introduce a new "selective" rewiring approach that rewires only graphs that based on their topology are likely to benefit. We further show that the optimal GNN depth varies between the graphs in a dataset and depends on the spectrum of the input graph; hence, aligning spectra across graphs allows for choosing a GNN depth that is close to optimal across the data set. We illustrate the utility of both intervention techniques in experiments on several common graph benchmarks.

## 1.1 RELATED WORK

**Performance heterogeneity** The interplay of model and data characteristics in graph learning was previously studied by Li et al. (2023); Liang et al. (2023), albeit only for node-level tasks. Li et al. (2023) establish a link between a graph's topological characteristics and performance heterogeneity. To the best of our knowledge, no such study has been conducted for graph-level tasks. As we discuss below, performance heterogeneity is linked to model generalization. Size generalization in GNNs has been studied in (Yehudai et al., 2021; Maskey et al., 2022; Le & Jegelka, 2024). For a more comprehensive overview of generalization results for GNNs, see also (Jegelka, 2022). Beyond graph learning, performance heterogeneity has been studied through the lens of example difficulty (Kaplun et al.) by analyzing a model's performance on individual instances in the test data.

**Graph Rewiring** Several rewiring approaches have been studied in the context of mitigating over-smoothing and over-squashing effects, most of which are motivated by topological graph characteristics that can be used to characterize both effects. Notable examples include rewiring based on the spectrum of the Graph Laplacian (Karhadkar et al., 2023; Arnaiz-Rodríguez et al., 2022), discrete Ricci curvature (Topping et al., 2022; Nguyen et al., 2023; Fesser & Weber, 2024a), effective resistance (Black et al., 2023), and random sparsification (Rong et al., 2019). When applied in graph-level tasks, these preprocessing routines are applied to all input graphs. In contrast, Barbero et al. (2023) propose a rewiring approach that balances mitigating over-smoothing and over-squashing and preserving the structure of the input graph. A more nuanced analysis of the effectiveness of standard rewiring approaches has been given in (Tortorella & Micheli, 2022; Tori et al., 2024).

## 1.2 SUMMARY OF CONTRIBUTIONS

Our main contributions are as follows:

1. We introduce graph-level *heterogeneity profiles* for analyzing performance variations of GNNs on individual graphs in graph-level tasks. Our analysis suggests that both message-passing and transformer-based GNNs display performance heterogeneity in classification and regression tasks.

2. We provide evidence that topological properties alone cannot explain graph-level heterogeneity. Instead, we use the notion of the Tree Mover's Distance to establish a link between class-distance ratios and performance heterogeneity.

3. We use these insights to derive lessons for architecture choices and data preprocessing. Specifically, we show that the optimal GNN depth for individual graphs depends on their spectrum and can vary across the data set. We propose a selective rewiring approach that aligns the graphs' spectra. In addition, we propose a heuristic for the optimal network depth based on the graphs' Fiedler value.

# 2 BACKGROUND AND NOTATION

Following standard convention, we denote GNN input graphs as $G = (X, E)$ with node attributes $X \in \mathbb{R}^{|V| \times m}$ and edges $E \subseteq V \times V$, where $V$ is the set of vertices of $G$.

## 2.1 GRAPH NEURAL NETWORKS

**Message-Passing Graph Neural Networks** Message-Passing (MP) (Gori et al., 2005; Hamilton et al., 2017) has become one of the most popular learning paradigms in graph learning. Many state of the art GNN architectures, such as *GCN* (Kipf & Welling, 2017), *GIN* (Xu et al., 2018) and *GAT* (Veličković et al., 2018), implement MP by iteratively updating their nodes' representation based on the representations of their neighbors. Formally, let $\mathbf{x}_v^l$ denote the representation of node $v$ at layer $l$. Then the updated representation in layer $l + 1$ (i.e., after one MP iteration) is given by

$$\mathbf{x}_v^{l+1} = \phi_l\Big( \bigoplus_{p \in \mathcal{N}_v \cup \{v\}} \psi_l\left(\mathbf{x}_p^l\right) \Big) .$$

Here, $\psi_l$ denotes an aggregation function (e.g., averaging) defined on the neighborhood of the anchor node $v$, and $\phi_l$ an update function (usually an MLP with trainable parameters) that computes the updated node representation. We refer to the number of MP iterations as the *depth* of the GNN. Node representations are initialized by the node attributes in the input graph.

**Transformer-based Graph Neural Networks** Several transformer-based GNN (GT) architectures have been proposed as an alternative to MPGNNs (Müller et al., 2023). They consist of stacked blocks of multi-head attention layers followed by fully-connected feed-forward networks. Formally, a single attention head in layer $l$ computes node feature as

$$\text{Attn}(\mathbf{X}^l) := \text{softmax}\Big(\frac{\mathbf{Q}\mathbf{K}^T}{\sqrt{d_k}}\Big)\mathbf{V} .$$

Here, the matrices the matrices $\mathbf{Q} := \mathbf{X}^l \mathbf{W_Q}$, $\mathbf{K} := \mathbf{X}^l \mathbf{W_K}$, $\mathbf{V} := \mathbf{X}^l \mathbf{W_V}$ are linear projections of the node features; the softmax is applied row-wise. Multi-head attention concatenates several such single-attention heads and projects their output into the feature space of $\mathbf{X}^l$. Notable instances of GTs include Graphormer (Ying et al., 2021) and GraphGPS (Rampášek et al., 2022).

A more detailed description of the GNN architectures used in this study can be found in Appendix A.1.

**Graph Rewiring** The topology of the input graph(s) has significant influence on the training dynamics of GNNs. Two notable phenomena in this context are over-smoothing and over-squashing. *Over-smoothing* (Li et al., 2018) arises when the representations of dissimilar nodes become indistinguishable as the number of layers increases. In contrast, *over-squashing* (Alon & Yahav, 2021) is induced by "bottlenecks" in the information flow between distant nodes as the number of layers increases. Both effects can limit the GNN's ability to accurately encode long-range dependencies in the learned node representations, which can negatively impact downstream performance. *Graph rewiring* was introduced as a pre-processing routine for mitigating over-smoothing and over-squashing by perturbing the edges of the input graph(s). A plethora of rewiring approaches, which leverage a variety of topological graph characteristics, have been introduced. In this paper we consider two rewiring approaches: FOSR (Karhadkar et al., 2023), which leverages the spectrum of the Graph Laplacian, and BORF (Nguyen et al., 2023; Fesser & Weber, 2024a), which utilizes discrete Ricci curvature. We defer a more detailed description of both approaches to Appendix A.2.

## 2.2 Tree Mover's Distance

The Tree Mover's Distance (short: TMD) is a similarity measure on graphs that jointly evaluates feature and topological information Chuang & Jegelka (2022). Like an MPGNN, it views a graph as a set of computation trees. A node's computation tree is constructed by adding the node's neighbors to the tree level by level. TMD compares graphs by characterizing the similarity of their computation trees via hierarchical optimal transport: The similarity of two trees $T_v, T_u$ is computed by comparing their roots $f_v, f_u$ and then recursively comparing their subtrees. We provide a formal definition of the TMD in Appendix A.4.

## 3 Establishing Graph-level Heterogeneity

In this section, we establish the existence of performance heterogeneity in common graph classification and regression tasks in both message-passing and transformer-based GNNs. We further provide empirical evidence that topological features alone are not sufficient to explain these observations.

### 3.1 Heterogeneity Profiles

To analyze performance heterogeneity, we compute *heterogeneity profiles* that show the average individual test accuracy over 100 trials for each graph in the dataset. For each dataset considered, we apply a random train/val/test split of 50/25/25 percent. We train the model for 300 epochs on the training dataset and keep the model checkpoint with the highest validation accuracy (see Appendix D for additional training details and hyperparameter choices). We then record this model's error on each of the graphs in the test dataset in an external file. We repeat this procedure until each graph has been in the test dataset at least 100 times. For the graph classification tasks, we compute the average graph-level accuracy (higher is better, denoted as ↑), and for each regression task the graph-level MAE (lower is better, denoted as ↓).

### 3.2 Heterogeneity in MPNNs and GTs

Figure 1 shows heterogeneity profiles for two graph classification benchmarks (Enzymes and Proteins) and one graph regression task (Peptides-struct). Profiles are shown for GCN (blue), a message-passing architecture, and GraphGPS (orange), a transformer-based architecture.

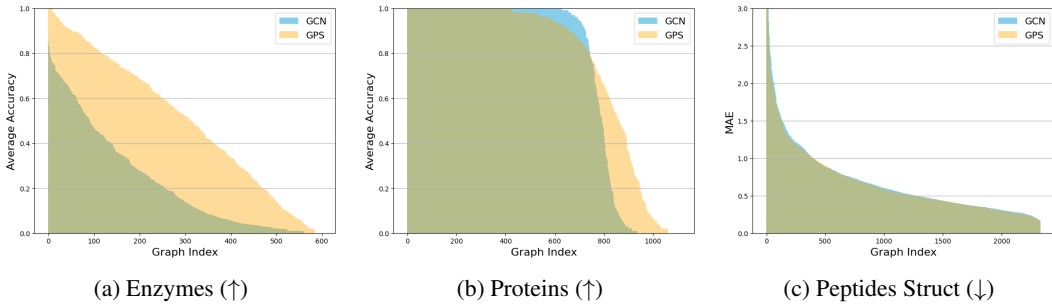

| (a) Enzymes (↑) | (b) Proteins (↑) | (c) Peptides Struct (↓) |

Figure 1: Comparison of heterogeneity profiles for message-passing (GCN) and transformer-based (GraphGPS) architectures.

For the classification tasks we observe a large number of graphs with an average GCN accuracy of 1, and a large number of graphs with an average accuracy of 0, especially in Proteins. This indicates that within the same dataset there exist some graphs, which GCN and GraphGPS always classify correctly, and others which they never classify correctly. Enzymes shows a similar overall trend: Some graphs have high average accuracies ($> 0.6$), while some are at or around zero. A comparable trend can be observed for GraphGPS, although the average accuracy here is visibly higher. Additional experiments on other graph classification benchmarks and using other MPGNN architectures confirm this observation (see Appendix B.1.2 and B.1.3). We refer to this phenomenon of large differences in graph-level accuracy within the same dataset as *performance heterogeneity*.

Furthermore, our results for Peptides Struct, a regression benchmark, indicate that heterogeneity is not limited to classification. The graph-level MAE of both GCN and GraphGPS varies widely between individual graphs in the Peptides-struct dataset. Appendix B.1.1 presents further experiments using Zinc, a regression dataset, with similar findings.

### 3.3 EXPLAINING PERFORMANCE HETEROGENEITY

As mentioned earlier, performance heterogeneity can also be found between individual nodes in node classification tasks (Li et al., 2023; Liang et al., 2023). They find that node-level heterogeneity can be explained well based on topological features alone: Nodes with higher degrees are provably easier to classify correctly than nodes with lower degrees Li et al. (2023). A natural question is whether this extends to graph-level tasks.

We test this with the following experiment, which is inspired by the analysis presented in Li et al. (2023) for node-level heterogeneity. For each dataset considered, we record the graph-level GCN accuracy averaged over 100 runs as above and compute several classical topological graph characteristics, such as assortativity, the clustering coefficient, and the spectral gap (see Appendix B.2 for details). We then train an MLP to predict the accuracy of a GCN on a given graph based on its topological features. While the MLP is able to fit the training data, it does not generalize well with test MSEs ranging from 0.1 to 0.27. Figure 11 in the Appendix visualizes this for the Enzymes dataset. Additional experiments in Appendix B.4 support the same message: Unlike in the node-level setting, topological features alone are insufficient to explain performance heterogeneity in the graph-level setting.

Next we evaluate potential drivers of heterogeneity using the Tree Mover's Distance (TMD) (Chuang & Jegelka, 2022), introduced above. Importantly, the TMD jointly evaluates topological and feature information, i.e., provides a richer data characterization.

**Definition 1 (Class-distance ratio)** *Let $\mathcal{D} = \{G_1, ..., G_n\}$ denote a graph dataset and $Y_i$ the correct label of the graph $G_i$. Using the TMD, we define the* class-distance ratio *of a graph $G_i$ as*

$$\rho(G_i) = \frac{\min_{G_j \in \mathcal{D} \setminus G_i}\{TMD(G_i, G_j) : Y_i = Y_j\}}{\min_{G_j \in \mathcal{D}}\{TMD(G_i, G_j) : Y_i \neq Y_j\}} .$$

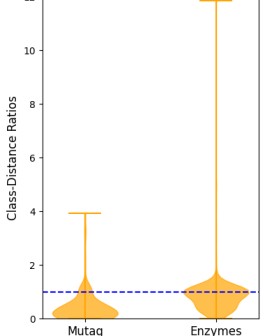

Figure 2: Class-distance ratios for graphs in Mutag and Enzymes.

In other words, we compute the TMD to the closest graph with the same label and divide it by the distance to the closest graph with a different label. If this ratio is less than one, we are closer to a graph of the correct label than to a graph of a wrong label. Figure 2 plots the class-distance ratios for all graphs in the Mutag and Enzymes datasets. (Due to the computational complexity of the TMD, we are unfortunately limited to analyzing rather small datasets). For both datasets, we can see that there exist graphs whose class-distance ratio is far larger than one, i.e., graphs which are several times closer to graphs of a wrong label than to graphs of their own label. Computing the Pearson correlation coefficient between a graph's class-distance ratio and its average GNN performance, we find a highly significant negative correlation for both datasets ($-0.441$ for Mutag and $-0.124$ for Enzymes). Graphs with a higher class-distance ratio have much lower average accuracies, i.e. are much harder to classify correctly. The following result on the TMD provides a possible explanation for this observation:

**Theorem 1 (Chuang & Jegelka (2022), Theorem 8)** *Given an $L$-layer GNN $h : \mathcal{X} \to \mathbb{R}$ and two graphs $G, G' \in \mathcal{D}$, we have*

$$\|h(G) - h(G')\| \leq \prod_{l=1}^{L+1} K_\phi^{(l)} \cdot TMD_w^{L+1}(G, G'),$$

*where $w(l) = \epsilon \cdot \frac{P_{L+1}^{l-1}}{P_{L+1}^l}$ for all $l \leq L$ and $P_L^l$ is the $l$-th number at level $L$ of Pascal's triangle.*

This result indicates that a GNN's prediction on a graph $G$ cannot diverge too far from its prediction on a similar graph $G'$, where similarity is defined via the TMD. A value of $\rho(G) > 1$ therefore

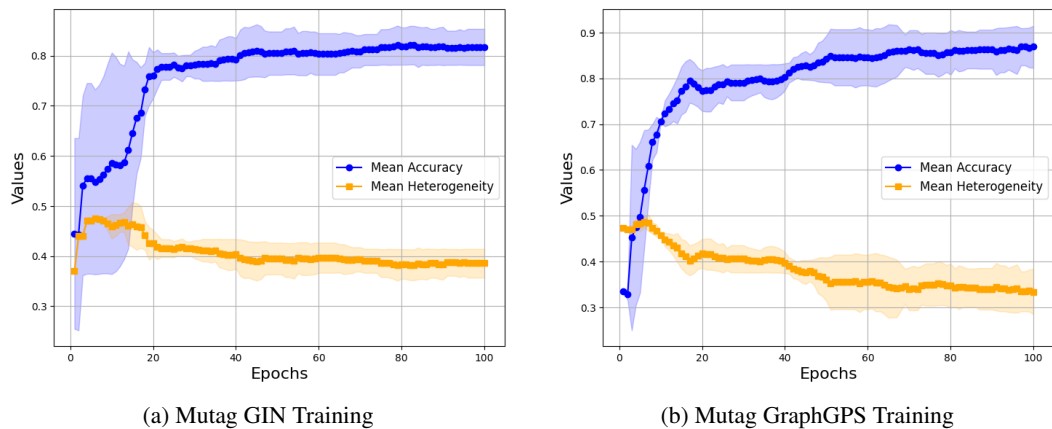

(a) Mutag GIN Training         (b) Mutag GraphGPS Training

Figure 3: Training performance comparison on the Mutag dataset using GIN (a MPGNN) and GraphGPS (a GT).

indicates that the GNN prediction cannot be too far from a prediction made on a graph with a different label, so graphs with $\rho(G) > 1$ are hard to classify correctly.

### 3.4 HETEROGENEITY APPEARS DURING GNN TRAINING

To better understand how performance heterogeneity arises during training, we analyze the predictions on individual graphs in the test dataset after each epoch. Figure 3 shows the mean accuracy and variance for GIN (message-passing) and GraphGPS (transformer-based) on Mutag. As we can see, both models have an initial increase in heterogeneity, followed by a steady decline over the rest of the training duration. Both models learn to correctly classify some graphs almost immediately, while others are only classified correctly much later. We should point out that these "difficult" graphs, i.e. graphs that are either learned during later epochs or never learned at all, are nearly identical for GIN and GraphGPS. GraphGPS, being a much more powerful model, learns to correctly classify more of these "difficult" graphs during later epochs. The (few) graphs which it cannot learn are also graphs on which GIN fails. Additional experiments on other datasets can be found in Appendix C.2. Across datasets and models, we witness this "learning in stages", where learning easy examples leads to an initial increase in heterogeneity, followed by a steady decrease.

## 4 HETEROGENEITY-INFORMED PRE-PROCESSING TECHNIQUES

In this section, we use heterogeneity profiles to show that preprocessing techniques can reinforce heterogeneity in graph-level learning. We focus primarily on rewiring techniques, but we also provide some experimental results with encodings in Appendix C.3. We find that while preprocessing methods such as rewiring are usually beneficial when averaged over all graphs, GNN performance on individual graphs can in fact drop by as much as 90 percent after rewiring. Deciding which graphs to preprocess is therefore crucial. We take a first step in this direction and propose a topology-aware selective rewiring method. As for encodings, our results indicate that investigating a similar, selective approach is a potentially fruitful direction (though unfortunately beyond the scope of this paper). We further note that existing theoretical results on improved expressivity when using encodings cannot explain the oftentimes detrimental effects observed in the heterogeneity analysis.

### 4.1 REWIRING HURTS (ALMOST) AS MUCH AS IT HELPS

For each dataset we compute heterogeneity profiles (based on graph-level accuracy over 100 runs) with and without rewiring, with a focus on two common rewiring techniques: BORF (Nguyen et al., 2023), a curvature-based rewiring method, and FoSR (Karhadkar et al., 2023), a spectral rewiring method. Figure 4 shows the results on the Enzymes and Proteins datasets with GCN. We find that while both rewiring approaches improve the average accuracy in each data set, the accuracy

for individual graphs can drop by as much as 95 percent. The graph-level changes in accuracy are particularly heterogeneous when using FoSR on both Enzymes and Proteins. BORF, while being less beneficial overall, also has less heterogeneity when considering individual graphs. This observation may indicate that not all graphs suffer from over-squashing or that over-squashing is not always harmful, providing additional evidence for arguments previously made by Tori et al. (2024).

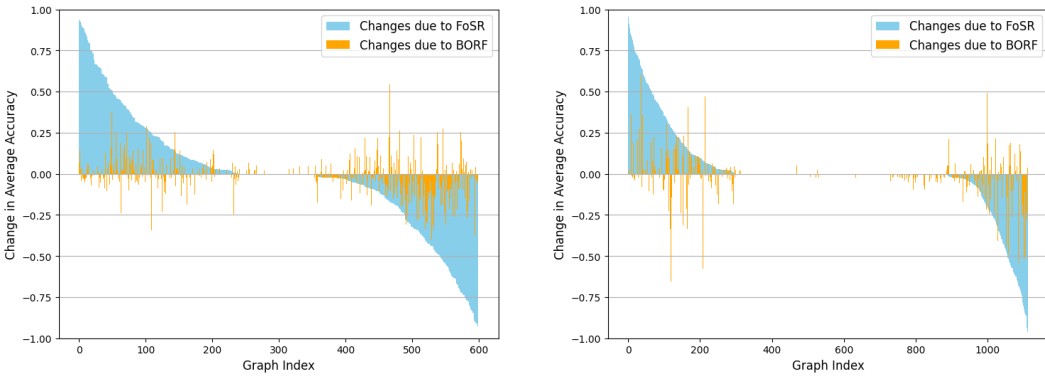

Figure 4: Comparison of BORF and FoSR methods applied to GCN on the Enzymes and Proteins datasets, sorted by FoSR changes from best to worst.

## 4.2 TOPOLOGY-AWARE SELECTIVE REWIRING

The large differences in performance benefits from FoSR on individual graphs reveal a fundamental shortcoming of almost all existing (spectral) rewiring methods: They rewire *all* graphs in a dataset, irrespective of whether individual graphs actually benefit. To overcome this limitation, we propose to rewire *selectively*, where a topological criterion is used to decide whether an individual graph's performance will benefit from rewiring. The resulting topology-aware selective rewiring chooses a threshold spectral gap $\lambda^*$ from the initial distribution of Fiedler values in a given dataset. Empirically, we find the median to work well. Using standard FoSR, we then add edges to all graphs whose spectral gap is below this threshold. The resulting approach, which we term *Selective-FOSR*, leaves graphs with an already large spectral gap unchanged. This "alignment" results in a larger degree of (spectral) similarity between the graphs in a dataset.

The effect on the spectral gaps of graphs in Enzymes and Proteins is plotted in Figure 5. Both datasets originally have a large number of graphs whose spectral gap is close to zero, i.e. which are almost disconnected and hence likely to suffer from over-squashing. Standard FoSR mitigates this, but simultaneously creates graphs with much larger spectral gaps than in the original dataset while more than doubling the spread of the distribution. Our selective rewiring approach avoids both of these undesirable effects. This also shows in significantly increased performance on all datasets considered when compared to standard FoSR, as can be seen in Table 1.

## 5 HETEROGENEITY-INFORMED MODEL SELECTION

So far, we have compared the heterogeneity profiles of different architectures, such as GCN and GraphGPS, on the same datasets, or compared profiles of datasets with and without rewiring. In this section, we now fix the preprocessing technique and base layer and focus on heterogeneity profiles at different GNN depths. This allows us to define a graph-level *optimal depth*. We show that the optimal depth varies widely within datasets, an observation that, we argue, is related to the graphs' spectral gap. We show that decreasing the variation in spectral gaps within a dataset via selective rewiring makes it easier to find a depth that works well on all graphs. At the same time, those insights motivate a heuristic for choosing the GNN depth in practise, which reduces the need for (potentially expensive) tuning of this crucial hyperparameter.

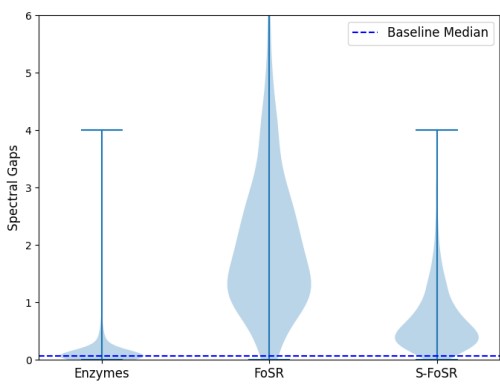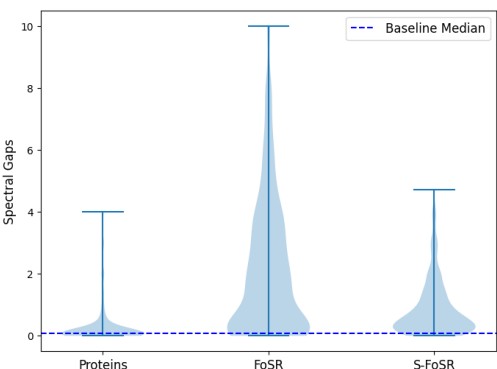

Figure 5: Spectral gap distributions in Enzymes and Proteins without any rewiring (left), with FoSR (center), and with Selective-FoSR (right).

| Model | Dataset | None | FoSR | S-FoSR |
|---|---|---|---|---|
| GCN | Peptides-struct ($\downarrow$) | $0.28 \pm 0.02$ | $0.27 \pm 0.01$ | $\mathbf{0.25 \pm 0.01}$ |
| | Peptides-func | $\mathbf{0.50 \pm 0.02}$ | $0.47 \pm 0.03$ | $0.48 \pm 0.02$ |
| | Enzymes | $23.8 \pm 1.3$ | $27.2 \pm 1.1$ | $\mathbf{30.1 \pm 1.0}$ |
| | Imdb | $49.7 \pm 1.0$ | $50.6 \pm 0.8$ | $\mathbf{51.2 \pm 1.1}$ |
| | Mutag | $72.4 \pm 2.1$ | $79.7 \pm 1.7$ | $\mathbf{81.5 \pm 1.4}$ |
| | Proteins | $69.9 \pm 1.0$ | $71.1 \pm 0.9$ | $\mathbf{72.6 \pm 1.1}$ |
| GIN | Peptides-struct ($\downarrow$) | $0.34 \pm 0.02$ | $0.27 \pm 0.02$ | $\mathbf{0.24 \pm 0.01}$ |
| | Peptides-func | $\mathbf{0.49 \pm 0.01}$ | $0.46 \pm 0.02$ | $0.49 \pm 0.02$ |
| | Enzymes | $27.1 \pm 1.6$ | $26.3 \pm 1.2$ | $\mathbf{28.6 \pm 1.3}$ |
| | Imdb | $68.1 \pm 0.9$ | $68.5 \pm 1.1$ | $\mathbf{69.0 \pm 0.9}$ |
| | Mutag | $81.9 \pm 1.4$ | $81.3 \pm 1.5$ | $\mathbf{84.9 \pm 1.0}$ |
| | Proteins | $71.3 \pm 0.7$ | $72.3 \pm 0.9$ | $\mathbf{72.6 \pm 0.8}$ |
| GAT | Peptides-struct ($\downarrow$) | $0.28 \pm 1.2$ | $0.29 \pm 0.01$ | $\mathbf{0.27 \pm 0.01}$ |
| | Peptides-func | $0.51 \pm 0.01$ | $0.49 \pm 0.01$ | $\mathbf{0.52 \pm 0.02}$ |
| | Enzymes | $23.8 \pm 1.2$ | $26.0 \pm 2.0$ | $\mathbf{31.5 \pm 1.8}$ |
| | Mutag | $70.2 \pm 1.3$ | $73.5 \pm 2.0$ | $\mathbf{78.5 \pm 1.7}$ |
| | Proteins | $71.3 \pm 0.9$ | $70.9 \pm 1.5$ | $\mathbf{72.5 \pm 0.8}$ |

Table 1: Classification accuracies of GCN with no rewiring, FoSR, and Selective FoSR using best hyperparameters. Highest accuracies on any given dataset and model are highlighted in bold. Unless specified otherwise, higher values are better.

## 5.1 OPTIMAL GNN DEPTH VARIES BETWEEN GRAPHS

Using the experimental setup for the heterogeneity profiles (see Section 3), we record the graph-level accuracy of GCNs with 2, 4, 6, and 8 layers respectively. We then record for each graph in the dataset the number of layers that resulted in the best performance on that graph and refer to this as the *optimal depth*.

**Optimal depth varies** Results on Enzymes and Peptides-struct are shown in Figure 6; additional results on other datasets and for other MPGNNs can be found in Appendix C.1. We observe a large degree of heterogeneity in the optimal depth of individual graphs. This is most striking in Peptides-Struct, where more than a quarter of all graphs attain their lowest MAE at only 2 layers, although there is also a substantial number of graphs that benefit from a much deeper network. The differences in average accuracy on a given graph between a 2-layer and an 8-layer GCN can be large: On Enzymes, we found differences of as much as 0.3. We provide the heterogeneity profiles for the 2, 4, 8 and 16-layer GCN trained on Enzymes in Appendix C.1. We observe that heterogeneity decreases with depth at the cost of accuracy. In particular, the 16-layer GCN trained on Enzymes does not classify *any* graphs correctly all the time, but it also consistently misclassifies only a small number of graphs. We believe that this is due to the model being much harder to train than the

shallower GCNs, possibly due to exploding/ vanishing gradients. Overall, our results indicate that choosing an optimal number of layers for a given dataset - usually done via grid-search - involves a trade-off between performance increases and decreases on individual graphs.

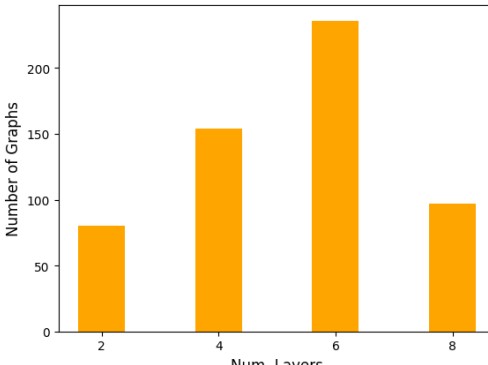 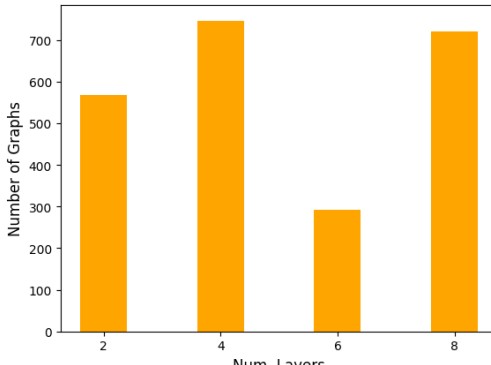

Figure 6: Distribution of graphs that attain their best GCN performance at 2, 4, 6, and 8 layers on Enzymes (left) and Peptides-struct (right).

**Analysis via Consensus Dynamics**   We believe that the heterogeneity in optimal depth can be explained using insights from average consensus dynamics, a classical tool from network science that has been used to study epidemics and opinion dynamics in social networks.

To define average consensus dynamics, we consider a connected graph $G$ with $|V| =: n$ nodes and adjacency matrix $A$. For simplicity, we assume that each node has a scalar-valued feature $x_i \in \mathbb{R}$, which is a function of time $t$. We denote these time-dependent node features as $\mathbf{x}(t) = (x_1(t), ..., x_n(t))^T$. The average consensus dynamics on such a graph are characterized by the autonomous differential equation

$$\dot{\mathbf{x}} = -L\mathbf{x},$$

which in coordinate form simply amounts to

$$\dot{x}_i = \sum_j A_{ij}(x_j - x_i) .$$

In other words, at each time step a node's features adapt to become more similar to those of its neighbours. For any given initialization $\mathbf{x_0} = \mathbf{x(0)}$, the differential equation will push the nodes' features towards a global "consensus" in which the features of all nodes are equal. Let $\hat{1}$ denote the vector of all ones. Since $\hat{1}^T \cdot L = 0$, $\hat{1}$ is an eigenvector of $L$ with zero eigenvalue, i.e.,

$$\hat{1}^T \cdot \dot{\mathbf{x}} = 0 \quad \Rightarrow \quad \hat{1}^T \cdot \mathbf{x} = \text{constant}.$$

Mathematically, this means that $x_i \to x^*$ for all $i$, as $t \to \infty$, where $\mathbf{x}^* = \frac{\hat{1}^T \cdot \mathbf{x_0}}{n}$ is the arithmetic average of the initial node features. Intuitively, these dynamics may be interpreted as an opinion formation process on a network of agents, who will in the absence of further inputs eventually agree on the same value, namely, the average opinion of their initial states. The rate of convergence is limited by the second smallest eigenvalue of $L$, the Fiedler value $\lambda_2$, with

$$\mathbf{x(t)} = \mathbf{x}^*\hat{1} + O(e^{-\lambda_2 t}).$$

We can think of the number of layers in a GNN as discrete time steps. Since graphs with a smaller Fiedler value take longer to converge to a global consensus, we would expect these networks to benefit from more GNN layers.

## 5.2 SPECTRAL ALIGNMENT ALLOWS FOR PRINCIPLED DEPTH CHOICES

Our discussion on average consensus dynamics suggests that individual graphs might require different GNN depths because approaching a global consensus takes about $1/\lambda_2$ time steps – in our case layers – where $\lambda_2$ is the graph's spectral gap. We also saw in the previous section that selective

(spectral) rewiring lifts the spectral gaps of (almost) all graphs in a dataset above a predetermined threshold $\lambda_2^*$ without creating graphs with very large spectral gaps. Motivated by these insights, we propose to use $1/\lambda_2^*$ as a heuristic for the GNN depth.

As Figure 7 shows, we empirically find that this approach works very well. On all three datasets depicted here (Mutag, Enzymes and Peptides-Struct), the ideal GCN depth turned out to be the integer closest to $1/\lambda_2^*$: 7 layers for both Mutag and Enzymes, and 12 layers for Peptides-struct. Using grid-search, we previously determined 4 layers to be optimal on both datasets after applying standard FoSR or no rewiring at all.

We now find that the accuracy obtained with a 7-layer GCN after applying selective FoSR is 3 percent higher than the accuracy obtained with selective FoSR and 4 layers on Mutag (5 percent on Enzymes). We also note that using 7 layers after applying FoSR or no rewiring is far from optimal on both datasets, highlighting the fact that the optimal depth changes with rewiring. Our observations on Peptides-struct are similar: Computing $1/\lambda_2^*$ suggests that we should use 12 layers on the selectively rewired dataset, which indeed turns out to be the ideal depth. This is far deeper than the 8 layers that are optimal with standard FoSR or no rewiring at all. Overall, we find that deeper networks generally perform better than shallow ones after selective rewiring. This might be surprising given that spectral rewiring methods are meant to improve the connectivity of the graph and hence allow for shallower models. We note that there is no such clear trend on the original datasets. For example, 8 layers perform almost as well on Mutag as 4. This further supports the notion that selective rewiring results in a spectral alignment between the graphs in a dataset.

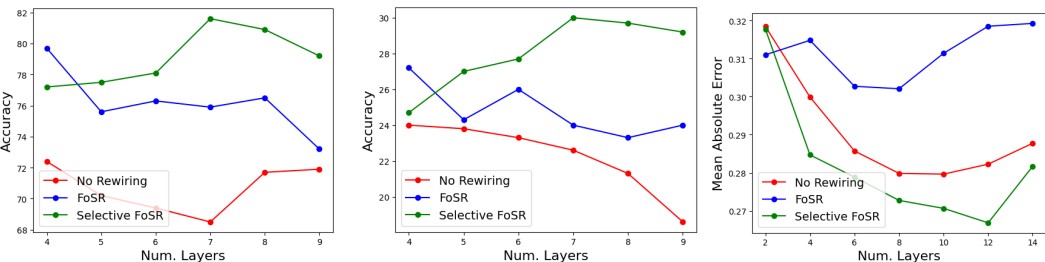

Figure 7: Depth vs accuracy/ MAE comparison for Mutag, Enzymes, and Peptides-Struct.

## 6 DISCUSSION

In this paper we analyzed performance heterogeneity in graph-level learning in both message-passing and transformer-based GNN architectures. Our results showed that unlike in the node-level setting, topological properties alone cannot explain graph-level heterogeneity. Instead, we identify large class-distance ratios as the main driver of heterogeneity within data sets. Our analysis suggests several lessons for architecture choice and data preprocessing, including a selective rewiring approach that optimizes the benefits of rewiring and a heuristic for choosing the optimal GNN depths. We corroborate our findings with computational experiments.

Our study of data preprocessing routines has focused on rewiring. However, recent literature has shown that encodings too can lead to significant performance gains in downstream tasks. While we include a preliminary study of heterogeneity in the context of encodings in Appendix C.3, a more detailed analysis is left for future work. We believe that a principled approach for selective encodings is a promising direction for extending the present work. Similarly, our theoretical understanding of common encodings such as LAPE and RWPE needs to be reassessed: Recent results argue that adding encodings makes GNNs more expressive, which does not explain the substantial detrimental effects on some graphs observed in our preliminary experiments. While our results suggest connections between performance heterogeneity and generalization, a detailed theoretical analysis of this link is beyond the scope of the present paper. We believe that in particular a detailed study of similarity measures in the style of the Tree Mover's Distance could provide valuable insights into transformer-based architectures. Lastly, this study aimed to work towards automating model choices in GNNs. We believe that the heterogeneity perspective could provide insights beyond GNN depth and preprocessing routines.

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

## A EXTENDED BACKGROUND

### A.1 MORE DETAILS ON GNN ARCHITECTURES

We provide a more detailed description of the GNN architectures considered in this paper.

**GCN** is a generalization of convolutional neural networks to graph-structured data. It learns a joint representation of information encoded in the features and the connectivity of the graph via message-passing. Formally, a GCN layer is defined as

$$\mathbf{H}^{(l+1)} = \sigma \left( \tilde{\mathbf{A}} \mathbf{H}^{(l)} \mathbf{W}^{(l)} \right) ,$$

where $\mathbf{H}^{(l)}$ denotes the node feature matrix at layer $l$, $\mathbf{W}^{(l)}$ the learnable weight matrix at layer $l$, $\tilde{\mathbf{A}} = \mathbf{D}^{-1/2} \mathbf{A} \mathbf{D}^{-1/2}$ the normalized adjacency matrix, $\mathbf{D}$ denoting the degree matrix and $\mathbf{A}$ the adjacency matrix. Common choices for the activation function $\sigma$ include ReLU or sigmoid functions.

**GIN** is an MPGNN designed to be as expressive as possible, in the sense that it can learn a larger class of structural functions compared to other MPGNNs such as GCN. GIN is based on the Weisfeiler-Lehman (WL) test for graph isomorphism, which is a heuristic for graph isomorphism testing, i.e., the task of determining if two graphs are topological identical. Formally, the GIN layer is defined as

$$\mathbf{h}_v^{(l+1)} = \text{MLP}^{(l)} \left( (1 + \epsilon) \, \mathbf{h}_v^{(l)} + \sum_{u \in \mathcal{N}(v)} \mathbf{h}_u^{(l)} \right) ,$$

where $\mathbf{h}_v^{(l)}$ is the feature of a node $v$ at layer $l$, $\mathcal{N}(v)$ is the set of neighbors of the node $v$, and $\epsilon$ is a learnable parameter. Here, the update function is implemented as a multi-layer perceptron $\text{MLP}(\cdot)$, i.e., a fully-connected neural network.

**GINE** is a variant of GIN that can take edge features $e_{i,j}$ into account. It is defined as

$$\mathbf{x}_i' = h_\Theta \left( (1 + \epsilon) \cdot \mathbf{x}_i + \sum_{j \in \mathcal{N}(i)} \text{ReLU}(\mathbf{x}_j + \mathbf{e}_{j,i}) \right) ,$$

where $h_\Theta$ is a neural network, usually an MLP.

**GAT** implements an attention mechanism in the graph learning setting, which allows the network to assign different importance (or weights) to different neighbors when aggregating information. This is inspired by attention mechanisms in transformer models and can enhance the representational power of GNNs by learning to focus on more important neighbors. The GAT layer is formally defined as

$$\mathbf{h}_v^{(l+1)} = \sigma \left( \sum_{u \in \mathcal{N}(v)} \alpha_{vu} \mathbf{W}^{(l)} \mathbf{h}_u^{(l)} \right) .$$

Here, $\alpha_{vu}$ denotes the attention coefficient between node $v$ and neighbor $u$, given by

$$\alpha_{vu} = \frac{\exp \left( \text{LeakyReLU} \left( \mathbf{a}^T \left[ \mathbf{W} \mathbf{h}_v^{(l)} || \mathbf{W} \mathbf{h}_u^{(l)} \right] \right) \right)}{\sum_{k \in \mathcal{N}(v)} \exp \left( \text{LeakyReLU} \left( \mathbf{a}^T \left[ \mathbf{W} \mathbf{h}_v^{(l)} || \mathbf{W} \mathbf{h}_k^{(l)} \right] \right) \right)} ,$$

where $\mathbf{a}$ is a learnable attention vector and $||$ denotes vector concatenation.

**GraphGPS** is a hybrid GT architecture that combines MPGNNs with transformer layers to capture both local and global information in graph learning. It enhances traditional GNNs by incorporating positional encodings (to provide a notion of node position) and structural encodings (to capture node-specific graph-theoretic properties). By alternating between GNN layers (for local neighborhood aggregation) and transformer layers (for global attention), GraphGPS can effectively learn both short-range and long-range dependencies within a graph. It uses multi-head attention, residual connections, and layer normalization to ensure stable and effective learning.

## A.2 Graph Rewiring

In this study we focus on two approaches that are representatives of the two most frequently considered classes of rewiring techniques.

**FOSR** Introduced by Karhadkar et al. (2023), this rewiring approach leverages a characterization of over-squashing effects using the spectrum of the Graph Laplacian. It adds synthetic edges to a given graph to expand its spectral gap which can mitigate over-squashing.

**BORF** Introduced by (Nguyen et al., 2023), BORF leverages a connection between discrete Ricci curvature and over-smoothing and over-squashing effects. Regions in the graph that suffer from over-smoothing have high curvature, where as edges that induce over-squashing have low curvature. BORF adds and removes edges to mitigate extremal curvature values. Fesser & Weber (2024a) show that the optimal number of edges relates to the *curvature gap*, a global curvature-based graph characteristic, providing a heuristic for choosing this hyperparameter.

## A.3 Positional and structural encodings

Structural (SE) and Positional (PE) encodings endow GNNs with structural information that they cannot learn on their own, but which is crucial for downstream performance. Encodings are often based on classical topological graph characteristics. Typical examples of positional encodings include spectral information, such as the eigenvectors of the Graph Laplacian (Dwivedi et al., 2023) or random-walk based node similarities (Dwivedi et al., 2022). Structural encodings include substructure counts (Bouritsas et al., 2022; Zhao et al., 2022), as well as graph characteristics or summary statistics that GNNs cannot learn on their own, e.g., its diameter, girth, the number of connected components (Loukas, 2019), or summary statistics of node degrees (Cai & Wang, 2018) or the Ricci curvature of its edges (Fesser & Weber, 2024b). The effectiveness of encodings has been demonstrated on numerous graph benchmarks and across message-passing and transformer-based architectures.

## A.4 More Details on the Tree Mover's Distance

For completeness, we recall the formal definition of the Tree Mover's Distance (TMD). We first define the previously mentioned notion of *computation trees*:

**Definition 2 (Computation Trees ((Chuang & Jegelka, 2022), Def. 1))** *For a graph G we recursively define trees $T_v^l$ ($T_v^1 = v$) as the depth-$l$ computation tree of node $v$. The depth-$(l+1)$ tree is constructed by adding the neighbors of the depth-$l$ leaf nodes to the tree.*

We further need the following definition:

**Definition 3 (Blank Tree ((Chuang & Jegelka, 2022), Def. 2))** *A blank tree $T_0$ is a tree with only one node whose features are given by the zero vector and with an empty edge set.*

Let $\mathcal{T}_u, \mathcal{T}_v$ denote two multisets of trees. If the multisets are not of the same size, it can be challenging to define optimal transport based similarity measures. To avoid this, we balance the data sets via augmentation with blank trees:

**Definition 4 (Blank Tree Augmentation ((Chuang & Jegelka, 2022), Def. 3))** *The function*

$$\rho : (\mathcal{T}_v, \mathcal{T}_u) \mapsto \left( \mathcal{T}_v \bigcup T_0^{\max(|\mathcal{T}_u| - |\mathcal{T}_v|, 0)}, \mathcal{T}_u \bigcup T_0^{\max(|\mathcal{T}_v| - |\mathcal{T}_u|, 0)} \right).$$

*augments a pair of trees with blank trees.*

We can now define a principled similarity measure on computation trees:

**Definition 5 (Tree Distance ((Chuang & Jegelka, 2022), Def. 4))** *Let $T_r, T_{r'}$ denote two trees with roots $r, r'$. We set*

$$\mathrm{TD}_w(T_r, T_{r'}) := \begin{cases} \|x_r - x_{r'}\| + w(L) \cdot \mathrm{OT}_{\mathrm{TD}_w}(\rho(\mathcal{T}_r, \mathcal{T}_{r'})) & \text{if } L > 1 \\ \|x_r - x_{r'}\| & \text{otherwise,} \end{cases}$$

*where $L = \max(\text{Depth}(T_r), \text{Depth}(T_{r'}))$ and $w : \mathbb{N} \to \mathbb{R}^+$ is a depth-dependent weighting function.*

Finally, we define a distance on graphs based on the hierarchical optimal transport encoded in the above defined tree distance:

**Definition 6 (TMD ((Chuang & Jegelka, 2022), Def. 5))** *Let $G, G'$ denote two graphs and $w, L$ as above. We set*

$$\text{TMD}_w^L(G, G') = \text{OT}_{\text{TD}_w}(\rho(\mathcal{T}_G^L, \mathcal{T}_{G'}^L)),$$

*where $\mathcal{T}_G^L$ and $\mathcal{T}_{G'}^L$ are multisets of the graph's depth-L computation trees.*

## B    HETEROGENEITY AND DATA CHARACTERISTICS

### B.1    ADDITIONAL HETEROGENEITY PROFILES

#### B.1.1    GCN

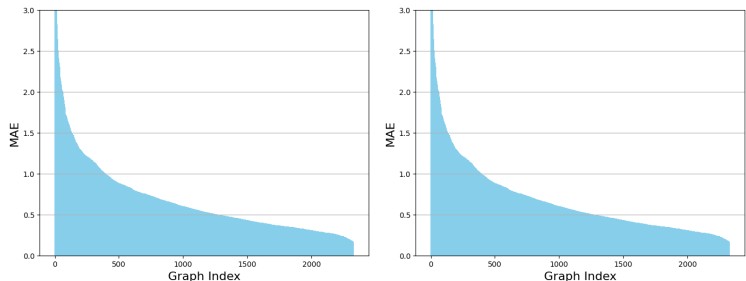

Figure 8: Heterogeneity profiles of a 4-layer GCN on the Zinc validation (left) and test datasets (right).

#### B.1.2    GIN

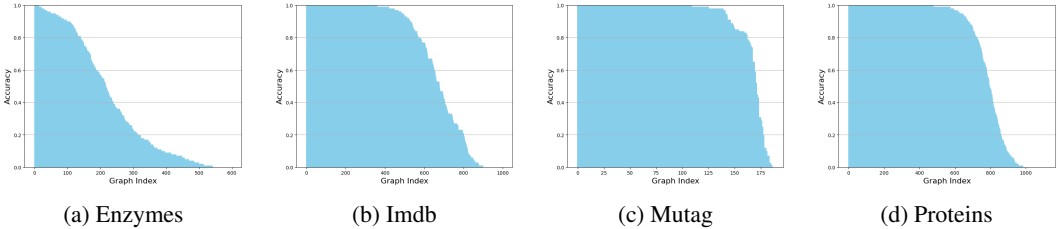

(a) Enzymes           (b) Imdb           (c) Mutag           (d) Proteins

Figure 9: Heterogeneity profiles obtained with GIN on Enzymes, Imdb, Mutag, and Proteins.

### B.1.3 GAT

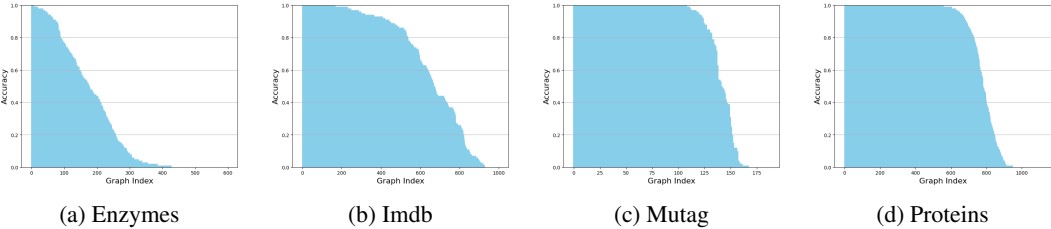

| (a) Enzymes | (b) Imdb | (c) Mutag | (d) Proteins |

Figure 10: Heterogeneity profiles obtained with GAT on Enzymes, Imdb, Mutag, and Proteins.

## B.2 TOPOLOGICAL PROPERTIES

Our MLP experiments in section 3.3 and sparse multivariate regression experiments in the appendix use the following topological properties of a graph $G = (V, E)$ with $|V| = n, |E| = m$ to predict average GNN accuracy on that graph.

- **Edge Density.** The edge density for an undirected graph is calculated as $\frac{2m}{n(n-1)}$, while for a directed graph, it is computed as $\frac{m}{n(n-1)}$.

- **Average Degree.** The average degree for an undirected graph is defined as $\frac{2m}{n}$, while for a directed graph, it is defined as $\frac{m}{n}$.

- **Degree Assortativity.** The degree assortativity is the average Pearson correlation coefficient of all pairs of connected nodes. It quantifies the tendency of nodes in a network to be connected to nodes with similar or dissimilar degrees and ranges between $-1$ and $1$.

- **Diameter.** In an undirected or directed graph, the diameter is the length of the longest shortest path between any two vertices.

- **Average Clustering Coefficient.** First define $T(u)$ as the number of triangles including node $u$, then the local clustering coefficient for node $u$ is calculated as $\frac{2}{\deg(u)(\deg(u)-1)}T(u)$ for an undirected graph, where $\deg(u)$ is the degree of node $u$. The average clustering coefficient is then defined as the average local clustering coefficient of all the nodes in the graph.

- **Transitivity.** The transitivity is defined as the fraction of all possible triangles present in the graph. Formally, it can be written as $3\frac{\text{Num. triangles}}{\text{Num. triads}}$, where a triad is a pair of two edges with a shared vertex.

- **Spectral Gap/ Algebraic Connectivity.** The *Laplacian matrix* $L$ of the graph is defined as: $L = D - A$ where $D$ is the degree matrix and $A$ is the adjacency matrix of the graph. The eigenvalues of the Laplacian matrix are real and non-negative, and can be ordered as follows:
$$0 = \lambda_1 \leq \lambda_2 \leq \cdots \leq \lambda_n$$
The *spectral gap* of the graph is defined as the second-smallest eigenvalue of the Laplacian matrix, i.e. $\lambda_2$. This value is also known as the *algebraic connectivity* of the graph, and it reflects the overall connectivity of the graph.

- **Curvature Gap.** Using Ollivier-Ricci curvature $\kappa(u,v)$ of an edge $(u,v) \in E$, we consider an edge to be *intra-community* if $\kappa(u,v) > 0$ and *inter-community* if $\kappa(u,v) < 0$. Following Gosztolai & Arnaudon (2021), we define the curvature gap as

$$\Delta\kappa := \frac{1}{\sigma}\left|\kappa_{\text{inter}} - \kappa_{\text{intra}}\right|$$

where $\sigma = \sqrt{\frac{1}{2}\left(\sigma_{\text{inter}}^2 + \sigma_{\text{intra}}^2\right)}$. The curvature gap can be interpreted as a geometric measure of how much community structure is present in a graph (Fesser & Weber, 2024a).

- **Relative Size of the Largest Clique.** The relative size of the largest clique is determined by calculating the ratio between the size of the largest clique in $G$ and $n$.

## B.3 MLP

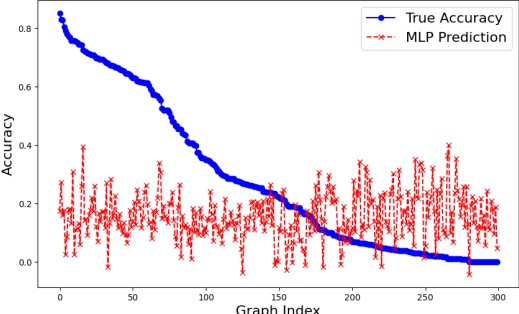

Figure 11: Accuracy of MLP predictions based on topological characteristics on Enzymes.

## B.4 SPARSE MULTIVARIATE REGRESSION RESULTS

| Features | Enzymes | Imdb | Mutag | Proteins |
|---|---|---|---|---|
| Edge Density | $4.151617e - 02$ | $2.898623e - 01$ | $-3.969576e + 00$ | $-2.560137e - 01$ |
| Average Degree | $6.882024e - 02$ | $2.217939e - 03$ | $1.999772e + 00$ | $3.252736e - 02$ |
| Degree Assortativity | $-3.918472e - 02$ | $-3.185174e - 01$ | $-8.107750e - 01$ | $4.863875e - 03$ |
| Diameter | $2.235096e - 03$ | $-6.465376e - 01$ | $9.388648e - 05$ | $3.698471e - 04$ |
| Average Clustering Coefficient | $-2.483046e - 01$ | $-4.472245e - 03$ | $4.440892e - 16$ | $-2.026657e - 01$ |
| Transitivity | $2.299293e - 01$ | $-7.803414e - 01$ | $0$ | $2.238747e - 01$ |
| Algebraic Connectivity | $-2.117908e - 02$ | $-2.505663e - 02$ | $4.693080e + 00$ | $1.093306e - 01$ |
| Curvature Gap | $-3.988714e - 07$ | $-5.182994e - 07$ | $-5.617996e - 04$ | $2.957293e - 08$ |
| Rel. Size largest Clique | $2.773373e - 02$ | $4.727640e - 03$ | $0$ | $9.840849e - 03$ |
| $R^2$ | $0.01670$ | $0.04559$ | $0.39944$ | $0.01626$ |

Table 2: Multivariate Sparse Regression coefficients for topological properties of graphs in the TU dataset. The target variable is the (normalized) graph-level GCN accuracy.

## C HETEROGENEITY AND MODEL CHARACTERISTICS

### C.1 ADDITIONAL RESULTS ON MODEL DEPTH

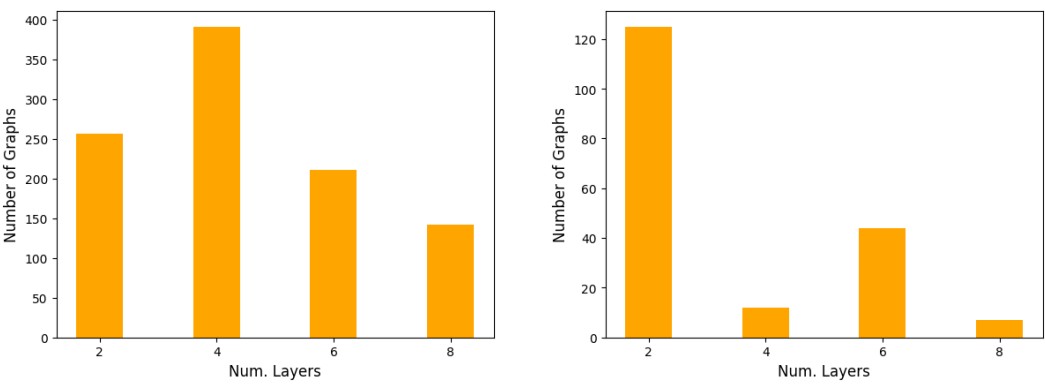

Figure 12: Distribution of graphs that attain their best GIN performance at 2, 4, 6, and 8 layers on Zinc (left) and Mutag (right).

### C.2 ADDITIONAL TRAINING DYNAMICS

### C.3 HETEROGENEITY PROFILES WITH ENCODINGS

For each dataset, we repeat the experimental setup described in section 4 without any encodings and with one common structural or positional encoding. Here, we consider the Local Degree

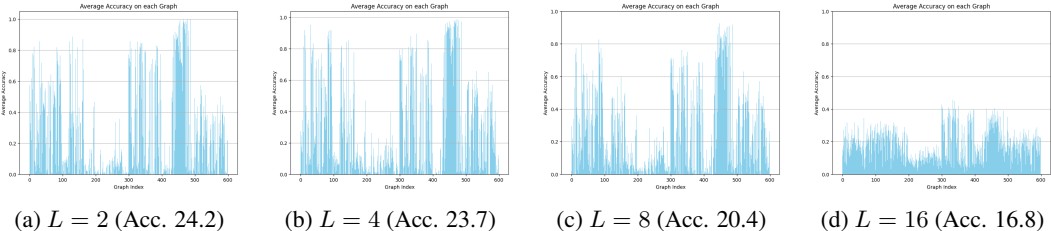

(a) $L = 2$ (Acc. 24.2)  (b) $L = 4$ (Acc. 23.7)  (c) $L = 8$ (Acc. 20.4)  (d) $L = 16$ (Acc. 16.8)

Figure 13: Comparison of GCN performance with different layer depths on the Enzymes dataset.

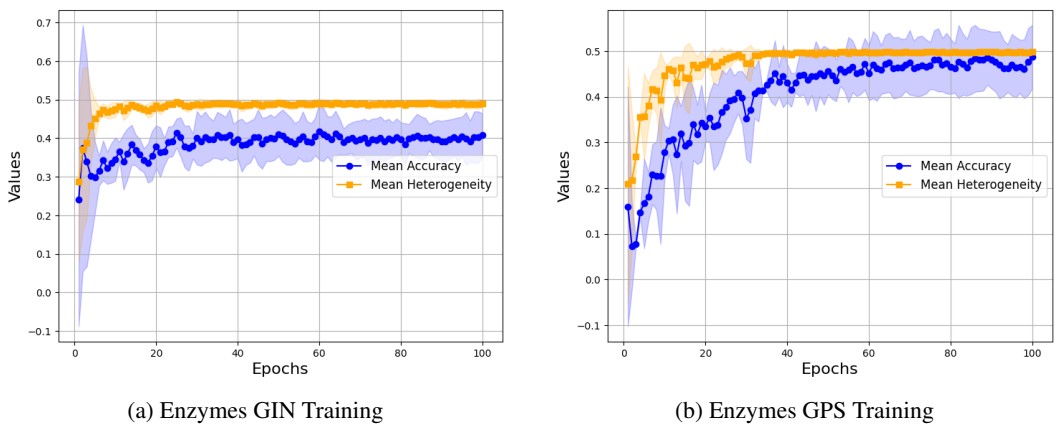

(a) Enzymes GIN Training        (b) Enzymes GPS Training

Figure 14: Training performance comparison on the Enzymes dataset using GIN and GPS.

Profile (LDP) (Cai & Wang, 2018), the Local Curvature Profile (LCP) (Fesser & Weber, 2024b), Laplacian-eigenvector positional encodings (LAPE) (Dwivedi et al., 2023), and Random-Walk positional encodings (RWPE) (Dwivedi et al., 2022). For each graph in a dataset, we compare the average graph-level accuracy over 100 runs with and without a given encoding. The results on the Proteins dataset with GCN are presented in Figure C.3. Note that values larger than one indicate that graph benefiting from a particular encoding, while values smaller than one indicate a drop in accuracy due to the encoding.

We can see that the structural encodings considered here, i.e., LDP and LCP, have especially large graph-level effects for GCN. Positional encodings such as LAPE or RWPE have much smaller effects on individual graphs. However, even for positional encodings, we find that there are always graphs on which GCN accuracy decreases once we add encodings. We note that 1) these detrimental effects have, to the best of our knowledge, not been reported in the literature, and that 2) they cannot be explained by existing theory. Encodings such as LCP, LAPE, and RWPE can be shown to increase the expressivity of MPNNs such as GCN and make them more powerful than the 1-Weisfeiler Lehman test. Drastic decreases in accuracy on individual graphs are therefore perhaps surprising.

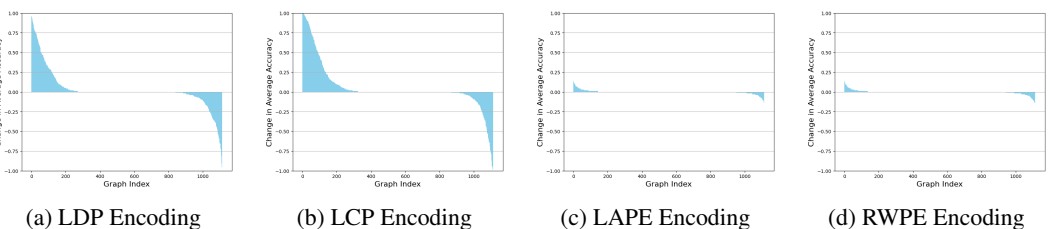

(a) LDP Encoding   (b) LCP Encoding   (c) LAPE Encoding   (d) RWPE Encoding

Figure 15: Different encodings applied to the GCN model on the Proteins dataset.

# D   HYPERPARAMETER CONFIGURATIONS

| Features | Enzymes | Imdb | Mutag | Proteins | Peptides-f | Peptides-s | Zinc |
|---|---|---|---|---|---|---|---|
| Layer Type | GCN | GCN | GCN | GCN | GCN | GCN | GINE |
| Num. Layers | 7 | 7 | 7 | 7 | 12 | 12 | 4 |
| Hidden Dim. | 64 | 64 | 64 | 64 | 235 | 235 | 64 |
| Learning Rate | 0.001 | 0.001 | 0.001 | 0.001 | 0.001 | 0.001 | 0.001 |
| Dropout | 0.1 | 0.1 | 0.1 | 0.1 | 0.1 | 0.1 | 0.1 |
| Batch Size | 50 | 50 | 50 | 50 | 50 | 50 | 50 |
| Epochs | 300 | 300 | 300 | 300 | 300 | 300 | 300 |
| Edges Added | 40 | 5 | 10 | 30 | 10 | 10 | 10 |

Table 3: Hyperparameter configurations used for experiments with selective-FoSR in the main text unless mentioned otherwise.

| Features | Enzymes | Imdb | Mutag | Proteins | Peptides-f | Peptides-s | Zinc |
|---|---|---|---|---|---|---|---|
| Layer Type | GCN | GCN | GCN | GCN | GCN | GCN | GINE |
| Num. Layers | 4 | 4 | 4 | 4 | 8 | 8 | 4 |
| Hidden Dim. | 64 | 64 | 64 | 64 | 235 | 235 | 64 |
| Learning Rate | 0.001 | 0.001 | 0.001 | 0.001 | 0.001 | 0.001 | 0.001 |
| Dropout | 0.1 | 0.1 | 0.1 | 0.1 | 0.1 | 0.1 | 0.1 |
| Batch Size | 50 | 50 | 50 | 50 | 50 | 50 | 50 |
| Epochs | 300 | 300 | 300 | 300 | 300 | 300 | 300 |
| Edges Added | 40 | 5 | 10 | 30 | 10 | 10 | 10 |

Table 4: Hyperparameter configurations used for experiments with FoSR in the main text unless mentioned otherwise.

| Features | Enzymes | Imdb | Mutag | Proteins | Peptides-f | Peptides-s | Zinc |
|---|---|---|---|---|---|---|---|
| Layer Type | GCN | GCN | GCN | GCN | GCN | GCN | GINE |
| Num. Layers | 4 | 4 | 4 | 4 | 8 | 8 | 4 |
| Hidden Dim. | 64 | 64 | 64 | 64 | 235 | 235 | 64 |
| Learning Rate | 0.001 | 0.001 | 0.001 | 0.001 | 0.001 | 0.001 | 0.001 |
| Dropout | 0.1 | 0.1 | 0.1 | 0.1 | 0.1 | 0.1 | 0.1 |
| Batch Size | 50 | 50 | 50 | 50 | 50 | 50 | 50 |
| Epochs | 300 | 300 | 300 | 300 | 300 | 300 | 300 |

Table 5: Hyperparameter configurations used for experiments without rewiring in the main text.

