# OpenReview forum: "Performance Heterogeneity in Message-Passing and Transformer-based Graph Neural Networks"
_ICLR.cc/2025/Conference — Submitted to ICLR 2025_

### Official Review · Reviewer_2jpS · 2024-10-28

**Soundness:** 1
**Presentation:** 1
**Contribution:** 1
**Rating:** 1
**Confidence:** 5

**Summary:**

The paper investigates the performance variability of GNNs in graph-level tasks.

**Strengths:**

Identifying the reasons behind the high variance in performance across different models is crucial; however, this work fails to offer new insights into this issue.

**Weaknesses:**

1. The paper highlights the variance in GNN performance on graph-level tasks; however, I believe this contribution does not offer new insights to the community. It is already well-established that Graph Neural Networks (GNNs) exhibit performance inconsistencies and that the effectiveness of different models often varies across datasets [1, 2].
Furthermore, the paper fails to propose a concrete solution to address this phenomenon. Inconsistencies in evaluation across sections further undermine its findings, as I will elaborate in my next comment.

2.
The authors investigate the inconsistency of performance on graph-level tasks by examining a range of models and datasets. However, the presentation lacks clarity regarding the consistency of their selections. For instance, in Section 3.2, three datasets are analyzed using two GNN architectures—specifically, GCN and GraphGPS. In contrast, Section 3.3 shifts to the Mutag dataset and the GIN model, raising questions about why different datasets and models were chosen for these sections.
Given that the paper's primary contribution aims to highlight an empirical phenomenon, I believe a more comprehensive evaluation is warranted, one that encompasses a broader array of benchmarks and various GNN architectures. Notably, the Mutag and Proteins datasets are recognized for their instability, as documented in previous studies [1]. The MUTAG dataset, in particular, is small and characterized by high variance, which has led to its declining usage in recent research.
Additionally, in Section 4, GPS is not tested at all; instead, GAT is utilized, despite all sections focusing on the same claim of heterogeneity in results. This inconsistency in model evaluation detracts from the paper’s coherence and impact.
The evaluation graph transformers focus solely on one type of graph transformer, GraphGPS, which is inadequate to substantiate the claim that “Our analysis suggests that both message-passing and transformer-based GNNs display performance heterogeneity in classification and regression tasks” (line 102). There exists a diverse range of graph transformers, as highlighted in studies such as [3, 4], which should be considered to strengthen the analysis.

3. The paper suffers from poor writing, featuring numerous grammatical errors and incomplete sentences. For instance, refer to lines 250, 352, 168, 576, and 187. Additionally, some sentences lack clear connections to the surrounding text, such as the statement: "Size generalization in GNNs has been studied in (Yehudai et al., 2021; Maskey et al., 2022; Le & Jegelka, 2024)."
The overall quality of English in the paper is inadequate and unprofessional, significantly detracting from the clarity and credibility of the research.

8. Overall, I find it difficult to see how the content of the paper supports the claims made in the abstract.

[1] A Fair Comparison of Graph Neural Networks for Graph Classification, Errica et al.
[2] Design Space for Graph Neural Networks, You et al., NeurIPS20.
[3] Do Transformers Really Perform Bad for Graph Representation?, Ying et al, 2021.
[4] Heterogeneous Graph Transformer, Hu et al., 2020.

**Questions:**

1. Can you explain the rationale behind the differing datasets and models used in Sections 3.2, 3.3. and 4?
2. While the paper highlights performance inconsistencies in GNNs, it does not present any concrete solutions to address this issue. What are your thoughts on proposing methodologies or frameworks to mitigate these inconsistencies in future work?

---

> ### Author Response · Authors · 2024-11-19
>
> We thank the reviewer for reading our submission. We politely disagree with the reviewer’s assessment of the paper. We believe that there are several misunderstandings regarding the motivation, methodology and contributions of our paper.
>
> We maintain that the paper’s focus on individual performance in graph-level learning offers a new perspective on performance heterogeneity. To the best of our knowledge, the selective rewiring approach and heuristic for the GNN depth are novel. This was also acknowledged by the other reviewers.
>
> We respond to a few of the points that the reviewer raised.
>
> > The paper highlights the variance in GNN performance on graph-level tasks; however, I believe this contribution does not offer new insights to the community. It is already well-established that Graph Neural Networks (GNNs) exhibit performance inconsistencies and that the effectiveness of different models often varies across datasets [1, 2]. Furthermore, the paper fails to propose a concrete solution to address this phenomenon. Inconsistencies in evaluation across sections further undermine its findings, as I will elaborate in my next comment.
> - We strongly disagree with the notion that there are no new insights to be gained here and believe that there is a misunderstanding. In fact, both of the papers pointed to by the reviewer focus on dataset-level accuracy (i.e. average over all nodes in node-level tasks and average over all graphs in graph-level tasks). They then study how model performance varies between datasets. This is fundamentally different from the graph-level approach we take: we focus on how performance varies **within** datasets, not **between** them. As such, our approach is similar to what [1] have proposed in the vision space. To the best of our knowledge, no comparable studies have been conducted for GNNs.
>
> > The authors investigate the inconsistency of performance on graph-level tasks by examining a range of models and datasets. However, the presentation lacks clarity regarding the consistency of their selections. For instance, in Section 3.2, three datasets are analyzed using two GNN architectures [...]
> - Many of the experiments are conducted on LRGB, which is a state-of-the-art graph learning benchmark. Some experiments involving the computation of the TMD are only performed on small data sets. As stated in section 3.3 this is because TMD does not scale to larger datasets.
> - We do not believe that testing rewiring methods with a global-attention based model like GPS would be insightful because the computation graph here is learned by the model (via attention) and is therefore not limited by the topology (unlike in message-passing). Our analysis based on consensus dynamics does therefore not immediately apply here. We would further like to point out that we are not aware of any papers in the literature that combine rewiring approaches with global-attention based models, most likely because those are usually assumed to not suffer from over-smoothing or over-squashing.
> - In the next version, we will extend our investigation of heterogeneity in section 2 to include additional transformer architectures.
>
> [1] Kaplun, Gal, et al. "Deconstructing Distributions: A Pointwise Framework of Learning." The Eleventh International Conference on Learning Representations.

---

### Official Review · Reviewer_wPWz · 2024-10-30

**Soundness:** 3
**Presentation:** 2
**Contribution:** 2
**Rating:** 3
**Confidence:** 4

**Summary:**

The paper investigates performance heterogeneity in graph neural networks (GNNs), specifically focusing on message-passing (MPGNNs) and transformer-based architectures. It addresses the challenges in understanding model performance variation on individual graphs within datasets used for graph-level learning. To capture performance variations, the authors introduce heterogeneity profiles and leverage the Tree Mover’s Distance (TMD) to demonstrate that both topological and feature information influence performance heterogeneity. The study explores how class-distance ratios, graph rewiring, and network depth impact heterogeneity, proposing a selective rewiring method and a depth-selection heuristic based on spectral alignment. The experiments validate these techniques, showing improved performance on multiple benchmarks.

**Strengths:**

- I think the attempted research question is fundamental and important for advancing GNN

- I do like the overall approach which is systematic

**Weaknesses:**

- My main concerns with respect to this paper are the contribution and novelty of the methodology and results. More specifically,

a. There is certainly merit in investigating the impact factors for the performance of GNN and I do like a more systematic approach. However, given the fact that GNN can be viewed as a function with the input of both structure and feature, it seems obvious that both feature and structure would affect the output/performance of GNNs. With this said, I am not convinced and not comfortable with the claim that topology is enough to explain node-level tasks (see [1] for an example).

b. One of the claimed contribution is the so-called "heterogeneity profiles". I do not see a detailed introduction or discussion on this technique and why it is novel. Based on the description on Section 3, it seems a standard random experiments/k-fold validation. Please correct me if I am wrong

c. The claimed research question investigates the factors that explain performance heterogeneity. However, the explained framework (tree mover distance) used in the paper is directly adopted from another paper. What is the new insight provided in this paper? In addition, there are many other distance metric such as FID can combine structure and feature. Why not consider those?

d. The paper tries to connect the experimental insight with graph rewiring and over-smoothing. The papers try to connect over-smoothing with the diffusion property of graph structure (fiddle eigenvalue of graph matrix). Despite being somewhat intuitive, it is not a strong explanation for over-smoothing as it is not clear how the diffusive property of graph structure would affect training or generalization. In addition, I think this diffusive property would largely affect node-level tasks. It is not entirely clear to me why this concept is applicable for graph-level tasks. Please explain. While I do think that selective graph rewiring could be a highlight of the paper, the paper does not go into detail in this regard. For example, how do you use the empirical result/theoretical result to obtain the proposed criteria for the selection?

- the presentation and organization of the paper need to be improved. I think the paper right now attempts to connect too many concepts and methods (performance heterogeneity, over-smoothing, graph rewiring e.t.c). I do admire the ambitious goal. However, in the current version of the paper, the connections among these concepts and methods are presented in a rather superficial way (this might be because of the page limit). I do encourage the authors to dive deeper into these connections as they are important for advancing GNN.


[1] "Subgroup generalization and fairness of graph neural networks." Advances in Neural Information Processing Systems 34 (2021): 1048-1061.

**Questions:**

see weakness

---

> ### Author Response · Authors · 2024-11-19
>
> We thank the reviewer for the feedback on our submission.
>
> > There is certainly merit in investigating the impact factors for the performance of GNN and I do like a more systematic approach. However, given the fact that GNN can be viewed as a function with the input of both structure and feature, it seems obvious that both feature and structure would affect the output/performance of GNNs. With this said, I am not convinced and not comfortable with the claim that topology is enough to explain node-level tasks (see [1] for an example).
> - We would like to point the reviewer to [1] and [2], which show experimentally that node degree is highly indicative of GNN performance in many real-world datasets. [2] can even rigorously connect node separability (as a proxy for GNN performance) and node degree in certain stylized settings.
>
> > One of the claimed contribution is the so-called "heterogeneity profiles". I do not see a detailed introduction or discussion on this technique and why it is novel. Based on the description on Section 3, it seems a standard random experiments/k-fold validation. Please correct me if I am wrong
> - We will provide a more explicit definition of heterogeneity profiles in the next version.
> - We believe that there is a misunderstanding with our methodology. This is not k-fold validation: k-fold validation would randomly split the data and then report average accuracy on the test set. Instead, we look at individual graph-level accuracy.
>
> > The claimed research question investigates the factors that explain performance heterogeneity. However, the explained framework (tree mover distance) used in the paper is directly adopted from another paper. What is the new insight provided in this paper? In addition, there are many other distance metric such as FID can combine structure and feature. Why not consider those?
> - Please note that while the TMD is indeed not novel (which is clearly stated in the paper), the notion of the class-distance ratio (CDR) is novel, as is the insight that CDR explains largely graph-level performance. We can think of the CDR as a notion of dataset separability derived from the TMD. Other graph distance measures could indeed also be used, but those generally scale even worse than TMD [3].
>
> > The paper tries to connect the experimental insight with graph rewiring and over-smoothing. The papers try to connect over-smoothing with the diffusion property of graph structure (fiddle eigenvalue of graph matrix). Despite being somewhat intuitive, it is not a strong explanation for over-smoothing as it is not clear how the diffusive property of graph structure would affect training or generalization. In addition, I think this diffusive property would largely affect node-level tasks. It is not entirely clear to me why this concept is applicable for graph-level tasks. Please explain. While I do think that selective graph rewiring could be a highlight of the paper, the paper does not go into detail in this regard. For example, how do you use the empirical result/theoretical result to obtain the proposed criteria for the selection?
> - Over-smoothing is not a focus of this paper, but rather model and hyperparameter choices that account for performance heterogeneity in graph-level learning. While rewiring is used to mitigate over-smoothing, the spectral rewiring method that we consider (FoSR) only address over-squashing, not over-smoothing.
>
> > the presentation and organization of the paper need to be improved. I think the paper right now attempts to connect too many concepts and methods (performance heterogeneity, over-smoothing, graph rewiring e.t.c). I do admire the ambitious goal. However, in the current version of the paper, the connections among these concepts and methods are presented in a rather superficial way (this might be because of the page limit). I do encourage the authors to dive deeper into these connections as they are important for advancing GNN.
> - We appreciate any feedback on improving the structure of the paper.
>
> [1] Liang, Langzhang, et al. "Resnorm: Tackling long-tailed degree distribution issue in graph neural networks via normalization." arXiv preprint arXiv:2206.08181 (2022).
>
> [2] Li, Ting Wei, Qiaozhu Mei, and Jiaqi Ma. "A metadata-driven approach to understand graph neural networks." Advances in Neural Information Processing Systems 36 (2024).
>
> [3] Chuang, Ching-Yao, and Stefanie Jegelka. "Tree mover's distance: Bridging graph metrics and stability of graph neural networks." Advances in Neural Information Processing Systems 35 (2022): 2944-2957.

---

> > ### Comment · Reviewer_wPWz · 2024-11-25
> >
> > I thank the author for the diligent response. However, many of my concerns remain. As such, I will keep the original score.
> >
> > Additional comment:
> >
> > There is a significant gap between "graph structure can be a good indicator or sufficient in style setting" and "graph structure is enough to explain node classification task". One straightforward example would be to use the contextual stochastic block model where the node identified from this model would depend on both structure and node feature.

---

### Official Review · Reviewer_pTz2 · 2024-11-02

**Soundness:** 3
**Presentation:** 3
**Contribution:** 3
**Rating:** 5
**Confidence:** 3

**Summary:**

This paper delves into the phenomenon of performance heterogeneity in graph-level learning, emphasizing how variations in graph topology influence model effectiveness across various architectures. Central to the study is the introduction of heterogeneity profiles, a novel analytical tool designed to systematically evaluate performance disparities across graph-level tasks. These profiles reveal that performance heterogeneity is shaped by factors beyond mere topological characteristics. Building on the analysis of heterogeneity profiles, the research progresses by proposing a selective rewiring strategy. This strategy aims to optimize network architecture based on the spectral properties of graphs, positing that aligning these properties can mitigate the need for extensive hyperparameter tuning by standardizing the optimal depth of GNN layers across different datasets.

**Strengths:**

1. The introduction of heterogeneity profiles as a tool in graph-level learning to analyze performance across graphs provides a new methodological avenue for studying GNNs.

2. The selective rewiring approach offers a pragmatic solution to a common problem in GNN deployment, potentially simplifying the model training process.

3. The experiments are well-designed, covering multiple datasets and configurations.

**Weaknesses:**

1. The paper does not provide a detailed description of how hyperparameters were tuned, only presenting the final hyperparameter results. Different hyperparameter settings, including adjustments to hidden dimensions, dropout rates, activation functions, and normalization techniques, could provide a stronger, more robust set of results.

2. The study lacks a detailed comparison with other state-of-the-art methods that aim to address similar challenges in GNNs. While the paper proposes innovative strategies for improving graph-level task performance, such as selective rewiring and optimized network depth based on heterogeneity profiles, it lacks empirical evidence showing that these approaches achieve state-of-the-art results on 1 or 2 benchmark datasets. This omission could undermine the perceived effectiveness and practical relevance of the proposed methods.

3. The datasets used in the study are relatively small in scale. Incorporating results from more extensive and challenging datasets, such as those from the OGB, would strengthen the validation of the techniques and enhance the paper’s impact.

**Questions:**

1. How might different settings for parameters affect the conclusions drawn from the study?

2. In line 298, these 'difficult' graphs are nearly identical for GIN and GraphGPS. Given that Figure 3 provides quantitative metrics but does not specify which exact graphs are considered "difficult" across both architectures, could the authors clarify how they determined that the same graphs pose difficulties for both GIN and GraphGPS?

---

> ### Author Response · Authors · 2024-11-19
>
> We thank the reviewer for the encouraging feedback.
>
> - We provide details on hyperparameter choices in the appendix. We will expand this description in the next version.
> - To the best of our knowledge, this work is the first to systematically study performance heterogeneity in graph-level learning. We would be grateful for any pointers to "other state-of-the-art methods" that the reviewer can provide.
> - LRGB is a commonly used, large-scale benchmark. We will add additional results on the suggested OGB tasks in the next version of the manuscript.
>
> Response to questions:
> - We find heterogeneity to appear with various widths, depths, learning rates, activation functions, and dropout rates.
> - We order the graphs by their indices in the datasets, allowing us to compare models.

---

> > ### Comment · Reviewer_pTz2 · 2024-12-03
> >
> > Thank you for providing a response to my questions. However, I did not see the new experimental results, nor did I see how different parameter settings might affect the conclusions of the study.

---

### Official Review · Reviewer_Z81b · 2024-11-04

**Soundness:** 3
**Presentation:** 4
**Contribution:** 3
**Rating:** 5
**Confidence:** 4

**Summary:**

This paper investigates the performance heterogeneity of message-passing and transformer-based architectures in graph-level tasks. Unlike previous studies that focused on node-level tasks, the authors find that graph topology alone does not fully explain heterogeneity. Instead, they establish a connection between class-distance ratios and performance heterogeneity using the Tree Mover's Distance. Building on these observations, the authors propose a selective rewiring approach and a heuristic for determining optimal GNN depths.

**Strengths:**

- Performance heterogeneity is a well-recognized and valuable research problem in graph learning.
- The proposed selective rewiring approach is promising for addressing performance heterogeneity in graph-level tasks.
- The observation that optimal network depth depends on the graph’s spectrum is intriguing, and the subsequent heuristic method for selecting the number of GNN layers is validated by the experimental results.

**Weaknesses:**

- The generalization capability of the proposed selective rewiring approach remains uncertain. While the motivation for this approach is empirically driven, the observations may be biased by the specific graph datasets tested. Would the conclusions hold on challenging open-source benchmarks, such as large-scale datasets in OGB (e.g., ogbg-ppa and ogbg-code2)?
- Given the proposed solutions, how can they be applied to new graph scenarios? For new graph datasets, is there a confidence measure for selective rewiring or heuristic GNN depth prediction?

**Questions:**

This work addresses an important yet challenging research topic in the graph field. I appreciate the focus on identifying performance heterogeneity in graph-level tasks. However, my primary concern is ensuring the proposed solution’s generalizability. For further details, please refer to the Weaknesses section.

---

> ### Author Response · Authors · 2024-11-19
>
> We thank the reviewer for the feedback and suggestions on how to improve the paper. Regarding the weakness that the reviewer mentions:
>
> - LRGB is the most common benchmark for testing how well models encode long-range interactions between nodes. It is generally considered a “challenging” benchmark. However, we will add additional results on the suggested OGB tasks in the next version of the manuscript.
> - The selective rewiring approach and depth heuristic are informed by the graph’s topology (characterized by the spectrum of the Graph Laplacian) and should be computed for the data set at hand. Our experimental results corroborate the utility of those heuristics.

---

### Comment · Area_Chair_bS8C · 2024-11-28

I would like to encourage the reviewers to engage with the author's replies if they have not already done so. At the very least, please
acknowledge that you have read the rebuttal.

---

### Meta-Review · Area_Chair_bS8C · 2024-12-19

**Metareview:**

The paper studies performance heterogeneity GNNs including MPNNs and transformer-based models, with a focus on graph-level tasks. It  introduces class-distance ratios (CDRs) derived from the Tree Mover’s Distance (TMD) to explain the heterogeneity. Two methods are proposed to address heterogeneity: selective graph rewiring and a heuristic for choosing GNN depth based on spectral properties. The results are on relatively small datasets, raising concerns about applicability to larger, more complex benchmarks like OGB. Overall, the experimental results are not convincing to the reviewers (missing baselines, inconsistencies in experimental setup). The presentation quality could be improved. For future work, the authors should expand the evaluation (larger benchmarks and baselines), and should make the experimental setup more consistent.

**Additional Comments On Reviewer Discussion:**

Reviewer Z81b highlighted concerns about the generalizability of the selective rewiring approach. The authors promised additional results in future revisions but did not provide immediate evidence during the rebuttal. Reviewer pTz2 requested more details and study of hyperparameters. While the authors provided some clarifications, the response lacked sufficient empirical additions to fully address the concerns. Reviewer 2jpS issued a strong rejection, citing the inconsistency in methodology and insufficient novelty. Reviewer wPWz also claims that many of their concerns remain. Overall, the response from the authors did not adequately address the concerns.

---

### Decision · Program_Chairs · 2025-01-22

Reject